# The yin–yang of kinase activation and unfolding explains the peculiarity of Val600 in the activation segment of BRAF

Christina Kiel[1,2,3]*, Hannah Benisty[1,2,3], Veronica Lloréns-Rico[1,2,3], Luis Serrano[1,2,3,4]*

[1]EMBL/CRG Systems Biology Research Unit, Centre for Genomic Regulation, Barcelona, Spain; [2]Universitat Pompeu Fabra, Barcelona, Spain; [3]Barcelona Institute of Science and Technology, Barcelona, Spain; [4]Institució Catalana de Recerca i Estudis Avançats, Barcelona, Spain

**Abstract** Many driver mutations in cancer are specific in that they occur at significantly higher rates than – presumably – functionally alternative mutations. For example, V600E in the BRAF hydrophobic activation segment (AS) pocket accounts for >95% of all kinase mutations. While many hypotheses tried to explain such significant mutation patterns, conclusive explanations are lacking. Here, we use experimental and in silico structure-energy statistical analyses, to elucidate why the V600E mutation, but no other mutation at this, or any other positions in BRAF's hydrophobic pocket, is predominant. We find that BRAF mutation frequencies depend on the equilibrium between the destabilization of the hydrophobic pocket, the overall folding energy, the activation of the kinase and the number of bases required to change the corresponding amino acid. Using a random forest classifier, we quantitatively dissected the parameters contributing to BRAF AS cancer frequencies. These findings can be applied to genome-wide association studies and prediction models.

*For correspondence: christina.kiel@crg.eu (CK); luis.serrano@crg.eu (LS)

**Competing interests:** The authors declare that no competing interests exist.

## Introduction

In a time of personalized medicine and tumor sequencing, determining which missense mutations affect disease phenotype and exploring the role of cellular and environmental context are crucial. In many oncogenes, mutations are enriched at specific amino acid positions ('mutation hotspots'), and it is not usually obvious if rare substitutions are passengers or disease-causing mutations. A striking example is the oncogenic serine/threonine kinase BRAF, for which the V600E mutation in its kinase activation segment (AS) accounts for >95% of all BRAF cancer mutations. BRAF is a serine/threonine protein kinase that is an upstream regulator of cellular responses such as cell division and differentiation and is mediated by the MEK/ERK signaling pathway (*Garnett and Marais, 2004*; *Wellbrock et al., 2004*). BRAF kinase is found mutated in both germline diseases (e.g. cardiofaciocutaneous and Noonan syndromes; (*Rauen, 2013*) and somatic cancers of the thyroid, skin, colon, and lung (*Holderfield et al., 2014*; *Ascierto et al., 2012*). BRAF contains an N-terminal region with a Ras-binding domain, which is followed by a cysteine-rich motif and a C-terminal kinase domain. BRAF is autoinhibited in a closed conformation by the interaction of the N-terminal conserved region 2 (following the Ras-binding and cysteine-rich domains) with the kinase domain, mediated by the interaction of two phosphorylated residues, Ser365 and Ser729, with a 14-3-3 dimer (*Figure 1—figure supplement 1*) (*Brummer et al., 2006*). Upon dephosphorylation of the N-terminal phosphorylated Ser365 by phosphatase PPII, the Ras-binding domain is free to interact with Ras at the plasma membrane. This releases autoinhibition and enables either homodimerization or heterodimerization

**eLife digest** Mutations in the gene that encodes a protein called BRAF are commonly found in certain cancers, such as melanomas. The same BRAF mutation is found in nearly all of these cancers. This mutation causes the 600th amino acid in the BRAF protein – an amino acid called a valine – to be replaced with another amino acid, a glutamate.

BRAF is a type of enzyme called a kinase, and it transmits signals inside cells to promote cell growth. Kinases work by adding a phosphate group to other proteins to alter their activity. The structure of the BRAF kinase contains a pocket-like shape, and the valine at position 600 sits buried inside this pocket when the enzyme is inactive. The "valine-to-glutamate" mutation (often called V600E for short) disrupts the interactions that create this pocket. This in turn results in a permanently active form of BRAF and uncontrolled cell growth. However, it remains unclear why the valine-to-glutamate mutation is so much more common in cancer cells than any other mutation that could affect the pocket in BRAF.

To address this question, Kiel et al. used a computational tool to generate three-dimensional models for all the different amino acid substitutions that could occur in BRAF's pocket. Each mutation was then assessed to see how it might destabilize the structure of BRAF. Only the mutations that affected the 600th amino acid were predicted to be able to open the pocket without destabilizing the part of the enzyme that adds phosphate groups to other proteins.

Kiel et al. validated their computational predictions by introducing normal or mutant versions of the BRAF-encoding gene into human cells grown in the laboratory. These experiments showed that a mutation that introduced an amino acid called histidine into position 600 could activate BRAF as much the valine-to-glutamate mutation. Kiel et al. suggest that this "valine-to-histidine" substitution is not found in cancers because it requires three changes to the DNA sequence of the BRAF gene, whereas the valine-to-glutamate substitution only requires one.

The results underscore the importance of considering changes at both the DNA and protein level when attempting to understand why certain cancer-causing mutations are more common than others.

with CRAF, ARAF, or KSR17; subsequent phosphorylation in the AS at Thr599 and Ser602 results in kinase activation (*Taylor and Kornev, 2011*; *Hmitou et al., 2007*; *Zhang and Guan, 2000*; for a recent review on the topic see *Lavoie and Therrien, 2015*).

Similar to other kinases, the BRAF kinase domain has two subdomains comprising a small N-terminal lobe and a large C-terminal lobe (*Figure 1A*) (*Scheeff and Bourne, 2005*; *Roskoski, 2010*). The N-terminal lobe contains the nucleotide-binding pocket and the phosphate-binding loop, while the C-terminal lobe binds the protein substrates and contains the catalytic loop. The two lobes, which are spatially connected through the AS, can move relative to each other in order to open or close the cleft. AS residues undergo hydrophobic interactions with the phosphate-binding loop and the 'αC helix' of the N-terminal lobe (making the 'hydrophobic pocket'), locking the kinase in its inactive state. In addition, the misalignment of spatially conserved hydrophobic residues in the N- and C terminal lobes ('hydrophobic spines') prevents catalytic activation (*Lavoie and Therrien, 2015*; *Hu et al., 2015*). Phosphorylation within the AS causes structural rearrangements of the AS, the αC helix and the phosphate-binding loop, reorienting the catalytic Asp of the DFG motif in a catalysis-competent orientation, thereby causing BRAF to become active.

There are two main hotspot regions for cancer-causing mutations in BRAF. Mutations in the phosphate-binding loop (residues 464 to 472) correspond to <1% of all BRAF mutations in cancer. The more important hotspot is found in the AS, with V600E being the most frequent BRAF somatic cancer mutation (98% in the COSMIC database) (*Supplementary file 1*; *Figure 1—figure supplement 2*) (*Cantwell-Dorris et al., 2011*; *Sarkozy et al., 2009*; *Holderfield et al., 2014*; *Lavoie and Therrien, 2015*). Less frequently found mutations at position Val600 are mutations to Asp, Lys, and Arg, which all require two nucleotide substitutions (*Davies et al., 2002*; *Lavoie and Therrien, 2015*). In the inactive conformation, Val600 is buried in a hydrophobic pocket made by residues from the N-terminal subdomain (Ala497, Phe498, Leu525, Leu485, Phe468, and Val487) and the AS (Leu597,

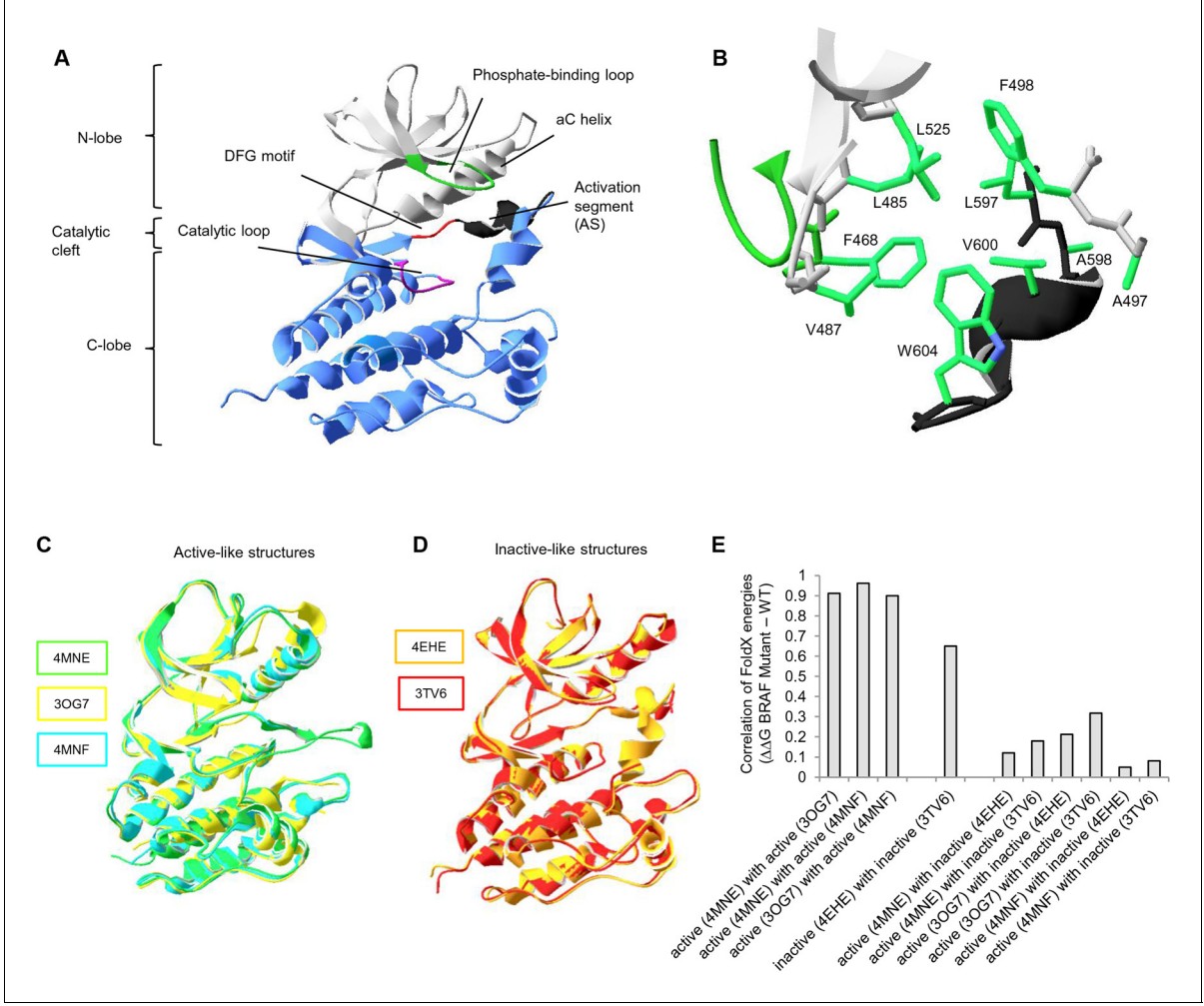

**Figure 1.** Overall structure of the kinase domain of BRAF, zoom into the hydrophobic pocket of BRAF, and active- and inactive-like BRAF kinase domain 3D structures used for structure-energy calculation. (**A**) Structure of the BRAF kinase, with functional regions indicated. The BRAF kinase domain has two subdomains, a small N-terminal lobe and a large C-terminal lobe. The small lobe contains the nucleotide-binding pocket and the phosphate-binding loop, while the large lobe binds the proteins substrates and contains the catalytic loop. The two lobes are spatially connected through the activation segment (AS) of the large lobe. Sequentially, the N- and C-terminal lobes are connected by the hinge, and the AS is part of the C-lobe that interacts with the N-lobe. Movement of the two lobes relative to each other opens and closes the cleft. (**B**) The hydrophobic pocket around amino acid Val600 represented using the backbone and side chain view. Backbone residues are colored according to their location in the protein (see *Figure 1A*). Specifically, Leu597, Ala598, Val600, and Trp604 of the AS together with, Phe468, Leu525, Leu485, Val487, Phe498, and Ala497 of the N-terminal subdomain build the hydrophobic pocket. All BRAF structural representations were done with SwissPdbViewer, using PDB entry 4EHE (chain B of the crystallographic unit). (**C**) Superimposition of active-like BRAF kinase structures. The structural representations were made using SwissPdbViewer (PDB entries 4MNE, 3OG7 and 4MNF). (**D**) Superimposition of inactive-like BRAF kinase structures. Structural representations were made using SwissPdbViewer (PDB entries 4EHE and 3TV6). (**E**) Pairwise correlation of FoldX energies for mutations in the hydrophobic pocket derived from active and inactive structures. Similar correlation results were obtained from FoldX energies using a recently published 3D structure of inactive monomeric BRAF (*Thevakumaran et al. (2015)*; PDB entry 4WO5, which is missing four residues in the AS/ data not shown).

The following figure supplements are available for figure 1:

**Figure supplement 1.** BRAF activation cycle.

**Figure supplement 2.** Cancer mutation frequencies in the hydrophobic pocket of BRAF.

**Figure supplement 3.** Basic principles of the FoldX force field, FoldX-based modeling, and the application of structure-energy calculations on mutations in BRAF's hydrophobic pocket.

Ala598, and Trp604) (*Figure 1B*). Substitution of this residue by charged amino acids (e.g. Glu) disrupts these interactions and results in constitutive kinase activation (*Wan et al., 2004*). BRAF V600E does not require RAF dimerization or interaction with Ras to be active (*Poulikakos et al., 2011*) yet has an increased propensity to form dimers (*Freeman et al., 2013*; *Roring et al., 2012*; *Thevakumaran et al., 2015*).

Whereas extensive research on BRAF in past years has provided enormous insight and understanding about the regulation of BRAF kinase and the abnormal activity of V600E (*Lavoie and Therrien, 2015*) no studies exist explaining why other amino acid substitutions in the hydrophobic pocket are not found with a high frequency in cancer. In principle, other mutations at the AS (such as Leu597 mutated into Glu), or in other parts of the hydrophobic pocket (e.g V487 into Glu or Leu525 into Glu), should also release the AS and cause constitutive kinase activation. Thus, to answer this question, we performed combined structure-energy, experimental and statistical analyses of mutations in the hydrophobic pocket. We show that V600E is the only single nucleotide substitution (Asp, Lys, and Arg, require two bases substitutions) that opens the AS through destabilization of autoinhibitory interactions, without significantly impairing the folding of the inactive or active kinase domain. We show that other mutations requiring three base substitutions (i.e. V600H) have kinase activities similar to V600E. We provide a quantitative measure for all parameters that contribute to BRAF cancer mutation frequencies by evaluating their importance using a random forest classifier. We anticipate that our results can be translated to other kinases and disease-causing proteins, provided that high-resolution X-ray structures are available.

## Results and discussion

### A quantitative measure for the destabilization of the hydrophobic pocket using structure-based energy calculations

Previous work on BRAF has shown that the V600E mutation is frequently found in cancer because it causes a disruption to the surrounding hydrophobic environment (*Wan et al., 2004*). To recapitulate what is already known in the literature and to have a quantitative measure for the destabilization of the hydrophobic pocket introduced by the V600E mutation, we used structure-based energy calculations. The protein design algorithm FoldX provides a quantitative estimation of the intermolecular forces and interactions contributing to the stability of proteins ($\triangle G$ = folding energy) based on high-resolution X-ray structures (*Figure 1—figure supplement 3A–B*) (*Guerois et al., 2002*; *Schymkowitz et al., 2005*; *Van Durme et al., 2011*). FoldX also enables amino acid replacements through side-chain rotamer modeling, allowing one to evaluate the energetic impact of a disease mutation on protein and/or complex stability (*Figure 1—figure supplement 3C–D*) (*Alibes et al., 2010*; *Pey et al., 2007*; *Rakoczy et al., 2011*; *Kiel and Serrano, 2014*). We performed FoldX-based molecular modeling of amino acid substitutions in the hydrophobic pocket of BRAF using active-like (4MNE (*Haling et al., 2014*), 3OG7 (*Bollag et al., 2010*) and the V600E mutant 4MNF (*Haling et al., 2014*)) and inactive-like (4EHE (*Mathieu et al., 2012*) and 3TV6 (*Wenglowsky et al., 2011*)) BRAF 'template' structures (*Figure 1C–D*; *Figure 1—figure supplement 3E*; *Supplementary file 1*). Using FoldX, we mutated every amino acid residue in the hydrophobic pocket of the five selected active and inactive structures to all amino acids, including itself (*Figure 1—figure supplement 3E*). This resulted in a total of 5 x 280 = 1400 structural models, and the change in folding energy ($\triangle\triangle G$ BRAF Mutant-WT) was determined (*Supplementary file 1*). Pairwise correlations of energies derived from active structures or inactive structures, respectively, show a good overall correlation (*Figure 1E*). In contrast, poor correlations were found when comparing energies from active and inactive structures, supporting the classification of the template structures.

All structural models with a change in FoldX energy ($\triangle\triangle G$ BRAF Mutant – WT) > 0.8 kcal were considered, as destabilizing mutants as this energy corresponds to a value twice the standard deviation of the energies calculated using the FoldX force field. To interpret the changes in FoldX energies, we needed to take into account several considerations (*Figure 1—figure supplement 3F*). First, mutations that destabilize the inactive conformation ($\triangle\triangle G$ BRAF_inactive) will drive the protein into a complex with chaperones (i.e. HSP90; *Grbovic et al., 2006*) and/or aggregation/degradation, thereby decreasing its overall effective concentration. Second, mutations that destabilize the active conformation ($\triangle\triangle G$ BRAF_active) will also result in the protein having a decreased effective

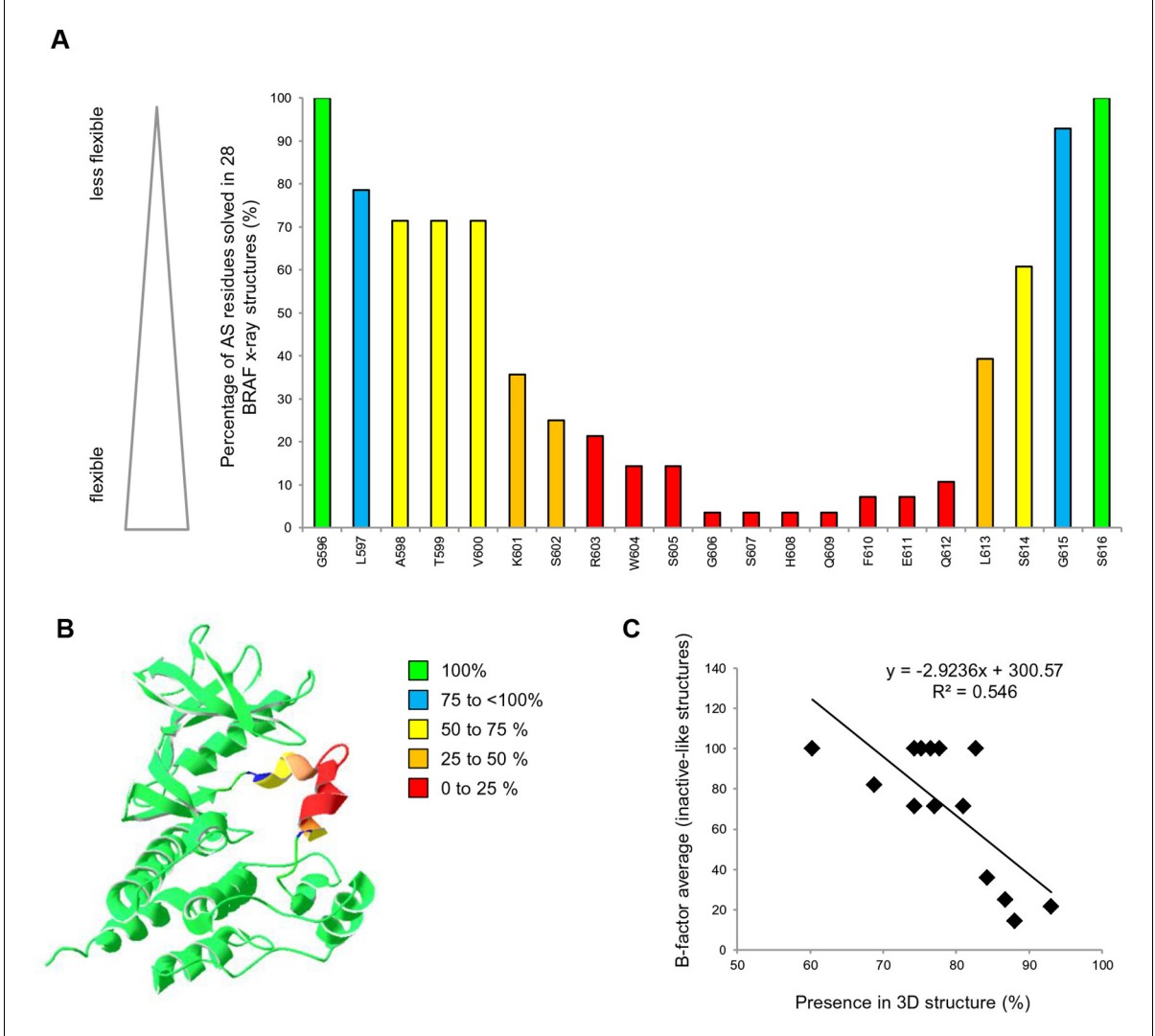

**Figure 2.** AS loop residues in 28 BRAF kinase structures and comparison with B-factors. (**A**) Percentage of the 28 BRAF X-ray structures that have a given AS residue solved. (**B**) Percentage of presence of AS loop amino acids in the X-ray structures, mapped onto a BRAF ribbon diagram (see legend for the color code). The structural representation was made using SwissPdbViewer (PDB entry 4EHE). (**C**) Normalized B-factor averages for loop residues from inactive structures (PDB entries 4EHE and 3TV6) plotted against the percentage of presence in the 28 BRAF X-ray structures.

concentration (unless they favor heterodimer formation and cause paradoxical activation (**Heidorn et al., 2010**; **Poulikakos et al., 2010**)). Third, unfavorable energy changes in the AS loop of the inactive structures will favor its release and therefore kinase activation ($\triangle\triangle$G BRAF_inactive_-loop). Structural inspection of 28 BRAF structures with different inhibitors showed that the AS loop between Leu597 and Gly615 is moderately to highly flexible (high B-factors) and consequently is unsolved in many structures (**Figure 2**; **Supplementary file 1**). Position Val600 is moderately flexible (70% solved in X-ray structures). This confirms previous predictions that the AS loop belongs to a region within the kinase domain (intra domain region) that has a large tendency to be disordered (**Lu et al., 2015**). Also, previous enhanced-sampling structure-based computational simulations proposed that the AS exhibited a significant tendency to switch from the ordered to unstructured conformation (**Marino et al., 2015**). Mutations in regions of high flexibility will have less impact on the unfolding of BRAF compared to those in conformational restricted regions. Thus, for the inactive state, we corrected the folding energies of the mutations in the AS loop ($\triangle\triangle$G BRAF_inactive) by the frequency for which the corresponding position is solved in the 28 crystal structures. This correction was not applied to the active-like structures because for these three active structures residues

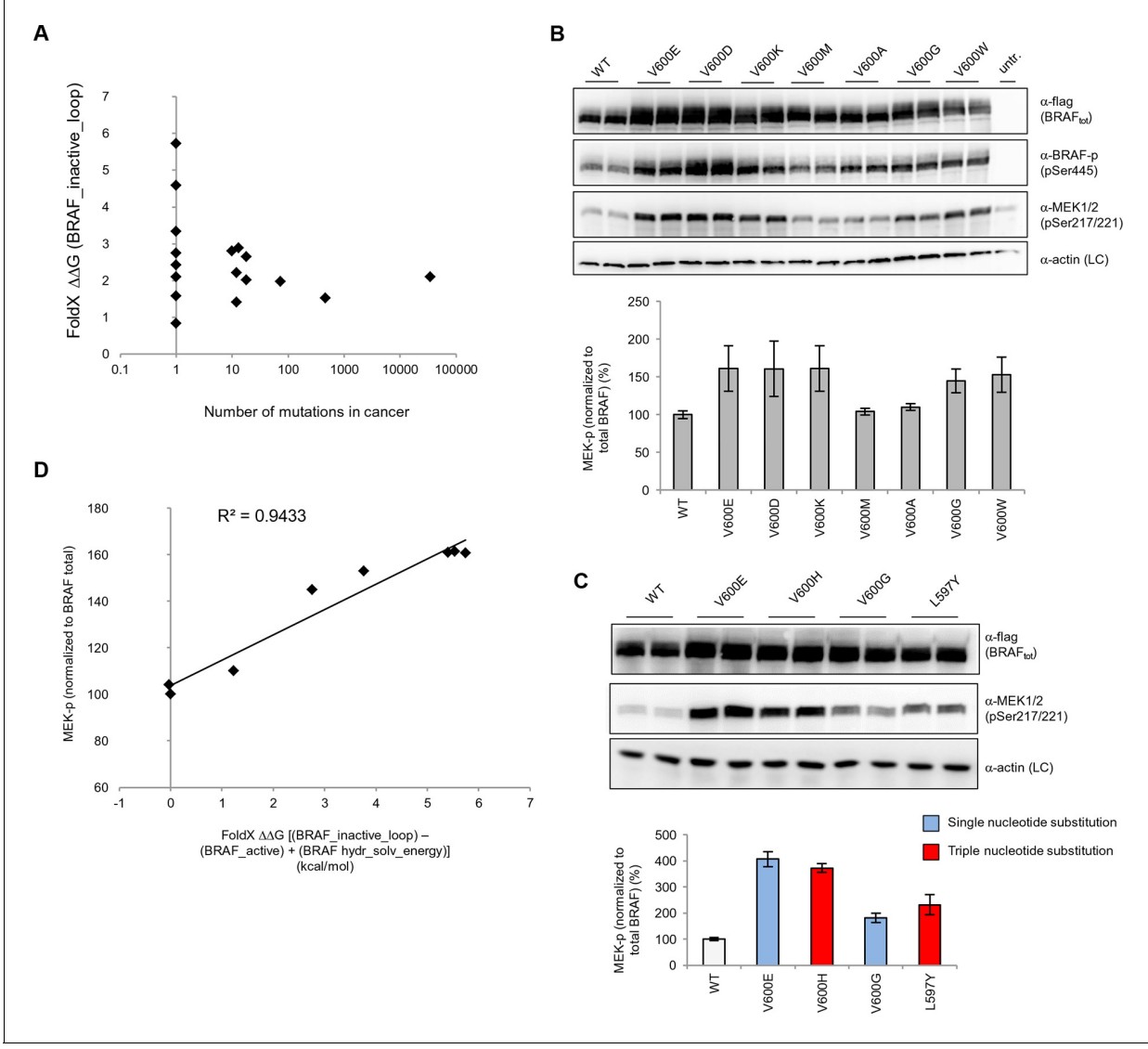

**Figure 3.** Structure-energy predictions and experimental analysis of mutations in the hydrophobic pocket of BRAF. (A) Comparison of the number of cancer mutations (>0) with destabilization of the hydrophobic pocket as predicted by FoldX (average energy values of 1EHE and 3TV6, 'FoldX △△G BRAF_inactive_loop'). (B) Representative Western blot (upper panel) for selected Val600 mutations expressed 24 hr in normal medium and quantified using ImageJ (lower panel). Two out of at least six biological replicates are shown. Bar graph shows the results of six biological replicates for the abundance of MEK-P normalized to total BRAF. (C) Representative Western blot (upper panel) analysis for selected single and triple nucleotide substitution BRAF mutations expressed 24 hr in normal medium and quantified using ImageJ (lower panel). Two out of at four biological replicates are shown. Bar graphs show the results of two biological and two technical replicates for the abundance of MEK-P normalized to total BRAF. (D) Correlation of FoldX energies with MEK phosphorylation normalized by the total BRAF levels. FoldX energies were calculated from the inactive loop energy [BRAF_inactive_loop] minus the FoldX energies derided from active structures [BRAF_active] plus the hydrophobic solvation energy as a factor in the FoldX force field [BRAF hydr_solv_energy].

The following figure supplements are available for figure 3:

**Figure supplement 1.** Mutations causing destabilization of the inactive loop and comparison with cancer frequencies.

**Figure supplement 2.** Mutations causing destabilization of the inactive structure above the threshold.

**Figure supplement 3.** Additional Western blots supporting *Figure 3B*.

**Figure supplement 4.** Additional Western blots supporting *Figure 3C*.

were solved only until position 600. The only available structure for which the loop had been solved (4MNE), had a high B-factor from position 601 onwards, but as there was no significant destabilization seen by FoldX, no correction was applied. After applying all these factors, we found several mutations that release the AS (FoldX energies above the threshold of 0.8 kcal/mol) and therefore could activate the kinase (*Supplementary file 1*).

## Integration of hydrophobic pocket destabilizing energies with nucleotide substitution frequencies and other biochemical parameters

The overall energy changes ($\triangle\triangle$G BRAF_inactive) as well as the energy changes in the AS loop alone ($\triangle\triangle$G BRAF_inactive_loop) that result from the introduction of mutations in the inactive structure, have very poor correlations with the occurrence of the corresponding mutation in tumors (*Figure 3A*; *Figure 3—figure supplement 1*; *Figure 3—figure supplement 2*). We suggest the following reasons for this:

1. As previously observed (*Davies et al., 2002*) the number of base substitutions required to change one residue into another amino acid is not the same for all mutations. For example, mutating Val600 to Glu requires only one base change, while mutating Val600 to Lys, Arg or Asp (other malignant mutations) requires two. Also, rare codons could result in lower protein expression levels.
2. The substitution of a hydrophobic residue into a bulkier hydrophobic one (i.e. Val to Trp), may be only moderately destabilizing because the flexible AS is expected to move slightly in order to accommodate and alleviate the van der Waals clashes. Thus, the AS may still be kept in a conformation closer to inactive-like depending on the chemical (hydrophobic) nature of the mutation.
3. Mutations could release the AS but at the same time destabilize the active or inactive conformations, driving the protein towards miss folding.
4. Mutations could mimic the phosphorylation of Ser602 and Thr599 at the AS (i.e. Ser into Asp, Thr into Glu), thus favoring the active conformation of the loop.

To see if these factors are responsible for the poor correlation between FoldX predicted energy changes and cancer frequency, we did a series of experiments and analyses described below.

## Experimental validation for interpreting the FoldX energies

We transiently expressed wild-type or mutant BRAF in HEK293 cells in normal growth medium and analyzed BRAF expression and the phosphorylation state of BRAF and MEK (*Figure 3B*; *Figure 3—figure supplement 3*). After correcting for differences in BRAF expression levels, we found that BRAF V600E phosphorylated MEK at higher levels than wild-type BRAF, and as predicted by FoldX, at similar levels to the double-nucleotide substitutions of V600D and V600K. In contrast, BRAF V600M and V600A yielded wild-type levels of MEK phosphorylation, suggesting that these are in fact passenger mutations (*Figure 3B*). The remaining mutations gave intermediate MEK phosphorylation levels. Indeed, the V600G mutation, which is also found in the germline and causes CFC syndrome (*Champion et al., 2011*) is an intermediate MEK activity mutant.

The fact that we found V600K and V600D mutants to be as active as the V600E mutant supports the hypothesis previously published (*Davies et al., 2002*) that the lower frequency of these mutants in cancer must be is due to the fact that two base substitutions are needed for changing Val600 into Lys or Asp, whereas only one is needed for V600E. We confirmed this further by identifying mutations that were not found in cancer (at positions 597 and 600 in the AS), that required three base changes, and that were predicted to be as activating as the most frequent cancer mutation found at these positions. Expression of these mutants (L597Y and V600H) in cells resulted in medium and high kinase activity as predicted (*Figure 3C*; *Figure 3—figure supplement 4*).

Replacement of V600 by bulkier hydrophobic residues (e.g. Met, Leu, Trp) resulted in weak (V600W) or no kinase activation. V600W, despite having a very high destabilizing AS loop FoldX energy in the inactive orientation, had a similar activity to that of V600G (*Figure 3B*). This supports our hypothesis that structural movements in the flexible AS could partially accommodate bulkier hydrophobic residues in the inactive orientation. Thus, we included the chemical nature/hydrophobicity as another factor. Considering the energies and parameters discussed above, we observed an excellent correlation between the FoldX predictions and MEK phosphorylation normalized by total BRAF (*Figure 3D*).

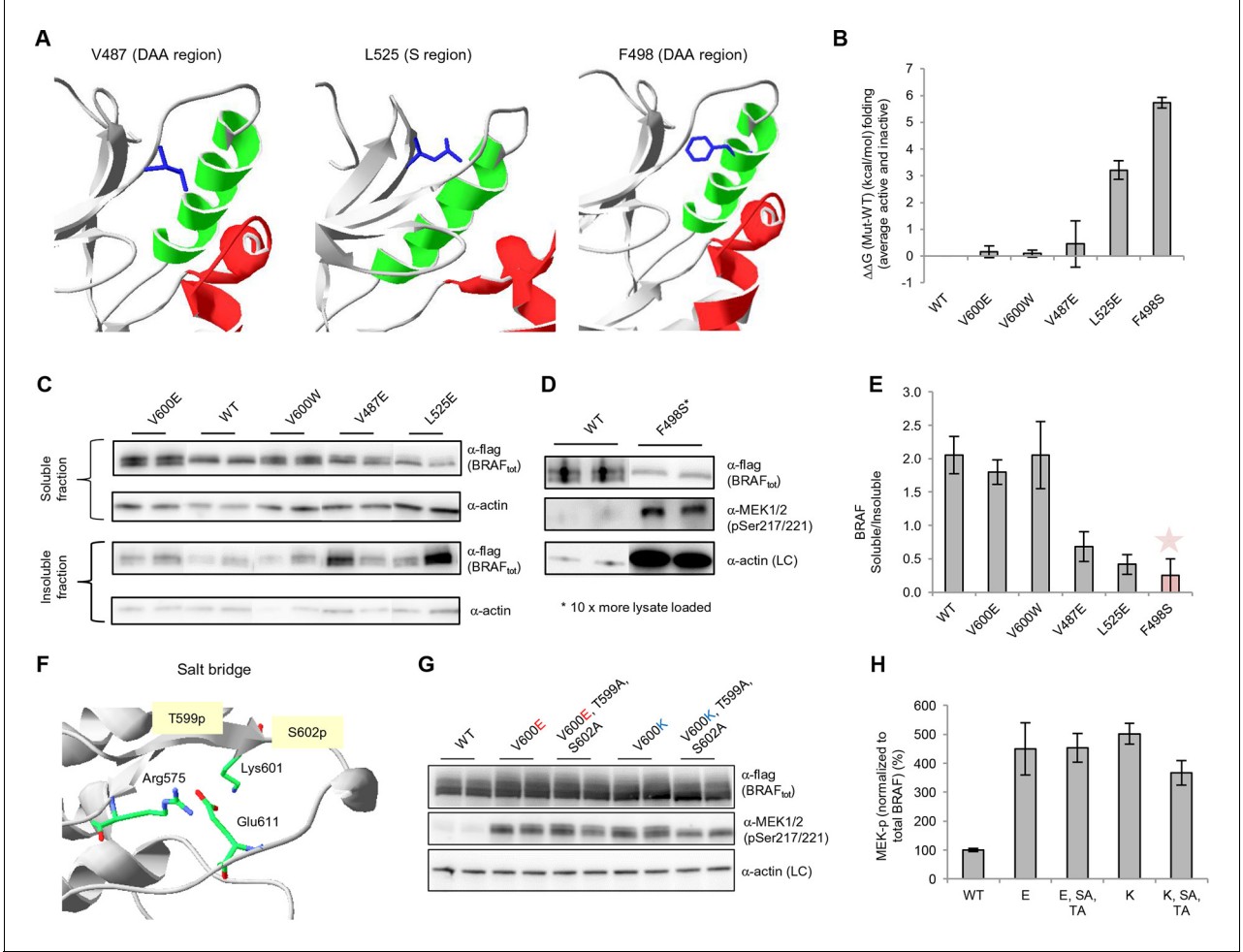

**Figure 4.** Structure-energy predictions and experimental analysis of mutations affecting the folding of BRAF and analysis of phosphorylation of Thr599 and Ser602 to keep the AS in a fixed active state. (**A**) Structural representations of the localization of Val487, Leu525, and Phe498 in BRAF (PDB entry 4EHE). (**B**) Destabilization of inactive and active states for V487E, L525E, and F498S BRAF (folding mutants) as predicted by FoldX. (**C**) Western blot analysis for BRAF mutations affecting folding. (**D**) Western blot analysis for BRAF F498S folding mutations. (**E**) Plot of BRAF soluble to insoluble ratios for the WT and mutations shown in the Western blots from pane (**C**) and (**D**), sorted in a similar order as in *Figure 3B*. Bar graphs show the results from two biological replicates. The soluble/insoluble value for BRAF F498S was estimated (see main text and represented with a star). (**F**) Illustration of the salt bridges that are proposed to stabilize the active conformation. The structural representation was done with the SwissPdbViewer, using PDB entry 4MNE. (**G**) Western blot analysis for the selected V600E and V600K mutations in combination with the T599A/S602A mutations expressed 24h in normal medium. (**H**) Quantifications of MEK phosphorylation levels normalized by total BRAF from (**G**) using ImageJ. Bars represent at least four biological replicates for the abundance of MEK-P normalized to total BRAF.

The following figure supplements are available for figure 4:

**Figure supplement 1.** Original western blots of spliced out lanes shown in *Figure 4C* and D.

**Figure supplement 2.** Comparing experimental protein solubility with FoldX predicted folding energies.

**Figure supplement 3.** MEK phosphorylation of wild-type and V600E, V487E, and L525E mutant BRAF in the supernatant.

**Figure supplement 4.** Conformations of Lys601 found in all structures having position 601 solved, and an overlay of ten active-like BRAF structures.

**Figure supplement 5.** Biological replicates in minimal (serum-free) growth medium.

**Figure supplement 6.** Analysis of the interactions in the BRAF RD motif, and expression levels of BRAF wild-type and the single V600E, E611A, and double V600E/E6111A mutants.

## Experimental analysis of mutations in the hydrophobic pocket predicted to disturb protein folding

Based on the data above, we can explain why V600E is the most frequent cancer mutation at position 600. We next wanted to analyze why no other mutation in the hydrophobic pocket - in a different position to Val600 - is found frequently mutated in cancer. Based on FoldX structure-energy calculations, we predicted that mutations in the hydrophobic pocket that destabilize the pocket and may thereby release the AS, would also affect the folding of the inactive and/or active kinase, thereby reducing the effective concentration and thus resulting in lower MEK phosphorylation (*Supplementary file 1*). We experimentally tested three mutations (V487E, L525E and F498S) that required one, or two (L525E) base changes (*Figure 4A–B*). F498S is predicted to be the most destabilizing, followed by L525E and V487E. By analyzing the soluble and insoluble fractions from transiently transfected HEK293 cells, we determined that the ratios between soluble and insoluble BRAF were similar for wild-type and the V600E and V600W mutants (*Figure 4C,E*; *Figure 4—figure supplement 1A*), while the V487E and L252E mutants resulted in significantly more insoluble protein. It is important to note that due to the very low levels of the BRAF F498S protein, it could only be detected when loaded in a 10-fold excess of lysate compared to wild-type (with no separation into soluble and insoluble fractions; *Figure 4D*; *Figure 4—figure supplement 1B*). Comparing the ratios of BRAF expressed in the soluble and insoluble fractions (*Figure 4E*) shows an inverse correlation with the folding energies as predicted by FoldX (*Figure 4B*; $R^2{\sim}0.67$, assuming for F498S a ratio of BRAF soluble/insoluble >0 and <0.4, ~0.2; *Figure 4—figure supplement 2*). However, despite the low levels of soluble F498S protein, it phosphorylated MEK at approximately the same level as wild-type BRAF (*Figure 4D*), while the V487E and L525E mutants, after normalizing by the total soluble protein gave higher MEK phosphorylation levels than wild-type (*Figure 4—figure supplement 3*).

Following the above analysis, we suggest that mutations that slightly destabilize both the folded conformation and the AS may cause small changes in ERK phosphorylation, which do not lead to cancer but may cause developmental defects. Indeed, three conservative RASopathy mutations are found in this region (L485F, L485S, and V487G; *Rauen, 2013*) (*Supplementary file 1*). Thus, other mutations in the pocket could indeed activate the kinase, but as a consequence of the resultant destabilization of the protein, they end up causing aggregation.

## Stabilization of the active conformation through salt bridges

Mutations that mimic phosphorylation can activate the kinase by interacting with Arg575, as shown for positions Thr599 and Ser602 (*Roskoski, 2010*). This is the mode of interaction for all so-called 'RD' kinases that become activated through phosphorylation within the activation segment (*Johnson and Lewis, 2001*). Structural inspection after superimposing all kinases suggests that mutating Lys601 to Glu could also lead to interaction with Arg575, with a small conformational change (*Figure 4F*; *Figure 4—figure supplement 4*). Thus, we added favorable energies to the Asp and Glu mutations made at those positions (the added energy value was determined by mutating phospho-Ser to Ser in the cAMP-dependent protein kinase structure (PDB entry 1ATP; *Zheng et al., 1993*; *Figure 4–figure supplement 5A*). This did not apply to position 600, however, which always points away from Arg575 in the active conformation, irrespective of if it is a Val or Glu, similar to the equivalent position in many other active kinases (*Figure 4—figure supplement 4*), and whose contribution to the active conformation energy is null. Finally, although in the V600E structure (PDB entry 4MNF) Glu600 forms a salt bridge with Lys507 in the αC helix (*Haling et al., 2014*), both residues are solvent exposed, and the Lys side chain is not structurally constrained. Therefore, the possible salt-bridge energy contribution is negligible (FoldX energy calculations suggest no energetic contribution between Glu600 and Lys507 ($\triangle\triangle$G (E600A) = 0.04+/- 0.1 kcal/mol). This explains why mutations to Lys or Arg are as activating as Glu and Asp.

As V600E or V600K does not stabilize the active conformation, we tested whether it still requires phosphorylation at Thr599 and Ser602 to keep the AS in a fixed active state (by interacting with Arg575) by mutating these residues to alanines (to disable phosphorylation). In normal growth medium, we observed either no change (V600E) or a slight reduction (V600K) in MEK phosphorylation (*Figure 4G–H*; *Figure 4—figure supplement 5*). This suggests that by opening the AS and preventing its closure, the kinase becomes active, independent of phosphorylation. These results were

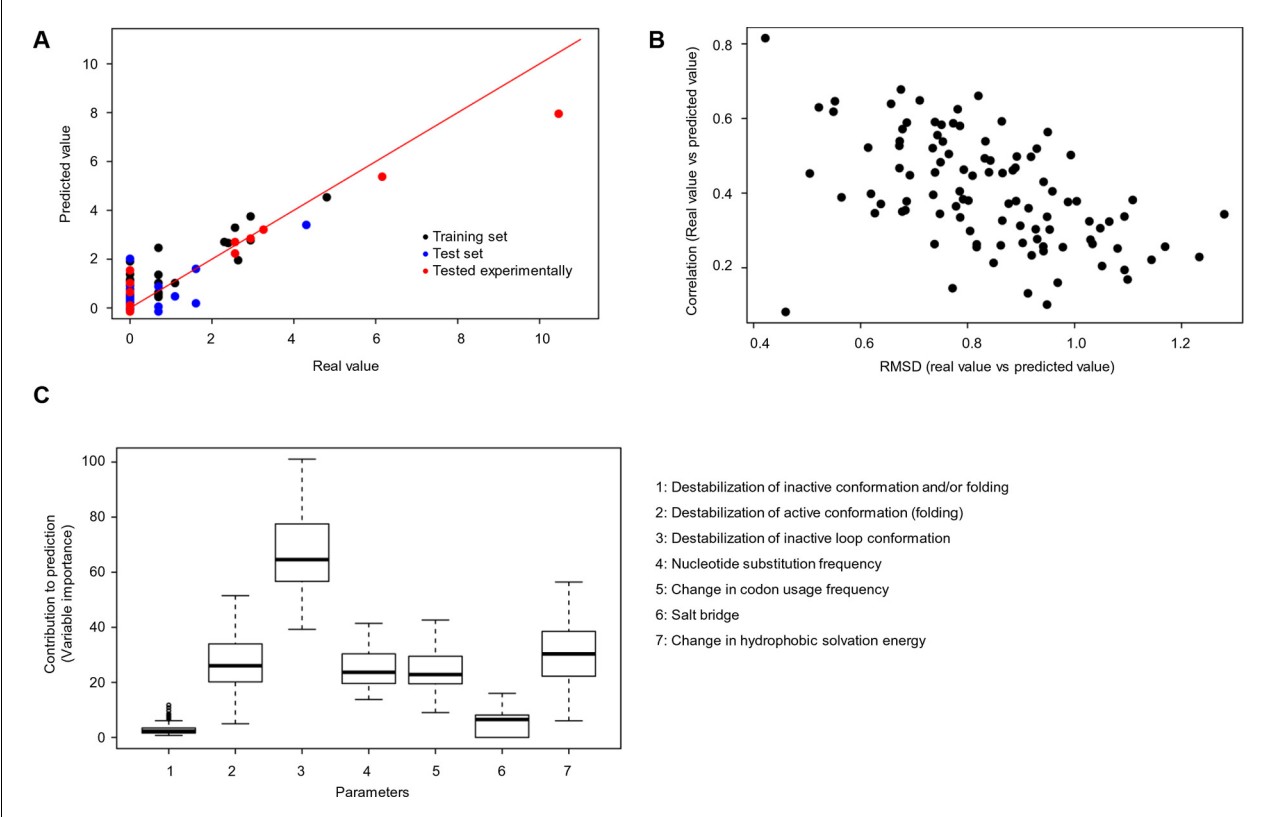

**Figure 5.** Quantitative contribution of individual factors to the prediction of cancer frequencies. (**A**) Comparison of real and predicted cancer frequencies (labelled 'real value' and 'predicted value') for one exemplary random forest prediction (run 16). Black dots represent mutations that were in the training set, blue dots the ones in the test set, and red are the mutations that were tested experimentally in this work (some of them were included in the training set, some of them in the test set). (**B**) Plot of RMSD against correlation for all individual random forest runs with V600E in the training set. The correlation is the correlation between the predicted value by the random forest ('predicted value') and the experimental value ('real value'), and the RMSD calculates the deviation of the predicted values from the real ones. (**C**) Results from random forest analyses with V600E in the training set. Abbreviation for parameters: 1) destabilization of inactive conformation and/or folding; 2) destabilization of active conformation (folding); 3) destabilization of inactive loop conformation; 4) nucleotide substitution frequency; 5) change in codon usage frequency; 6) salt bridge; and 7) change in hydrophobic solvation energy.

The following figure supplement is available for figure 5:

**Figure supplement 1.** Random forest analyses without V600E in the training set.

additionally supported by mutating Glu611 to Ala in the context of V600E. As Glu611 forms a salt bridge with Arg575, this interaction may partially stabilize the open conformation. However, as MEK phosphorylation did not change (*Figure 4G–H*; *Figure 4—figure supplement 6B–C*), it is more likely that this salt bridge contributes little or nothing to stabilization.

## Dissecting the contribution of individual parameters to the prediction of cancer frequencies

We used random forest predictions to analyze the quantitative contribution of individual factors to the prediction of cancer frequencies. In addition to the six parameters described above, we also included as a parameter the change in codon usage frequency due to a mutation (*Supplementary file 1*). If a frequent codon is mutated to a rare one, this could affect translation efficiency and protein levels (*Lampson et al., 2013*). To see if a combination of the factors discussed above can be used to predict the observed mutation frequency in cancer, we constructed a random forest classifier (*Figure 5A–B*). This ensemble learning technique identifies the contributions of individual 'trees' (here, FoldX energies, nucleotide substitutions, and codon frequencies) to an output

(here, cancer frequencies). As values not given in the training set cannot be extrapolated by the random forest method, we ran two sets of 100 predictions. For each prediction, we trained with a random subset of samples, using ~70% of the data and balancing mutations with low and high cancer frequencies. The V600E mutation was included in only one set, and the importance values for all seven parameters for all sets were kept. Next, we ran the trained random forest on the remaining ~30% of the data and calculated the root mean square deviation (RMSD) as well as the correlation between the real data and the predicted values. The ratio of this correlation to the RMSD was us as a performance indicator for each run. The importance values of the seven different parameters were similar between sets, suggesting that the presence or absence of V600E did not affect the training outcome (*Figure 5C*; *Figure 5—figure supplement 1*). The AS loop energy was the highest contributor to the random forest prediction of cancer frequencies (parameter 3; ~70% ), while parameters 2 (folding energy active conformation), 4 (nucleotide substitution frequency), 5 (change in codon usage frequency), and 7 (hydrophobic solvation energy) contributed almost equally, and 1 and 6 (destabilization of inactive conformation and the salt bridge) had very little contribution (*Figure 5C*). Ensemble methods, such as random forests, have several advantages compared to non-ensemble machine-learning methods, such as better handling of small sample sizes and high dimensionality, increased robustness and limited overfitting. The contributions of the different features calculated in this study are quite robust, and in the two cases analyzed (with and without V600E in the training set), they were found to be comparable and to follow the same order.

## Conclusions

We provide a complete picture for the genotype-phenotype associations of the hydrophobic pocket of the BRAF kinase domain and emphasize the importance of a balance between increased activity and loss in stability and/or folding. By using structure-energy calculations and a number of nucleotide substitutions, we were able to reconcile why V600E is by far the most frequent cancer mutation. We show here, that the effect of a mutation on folding depends on the structural flexibility of the

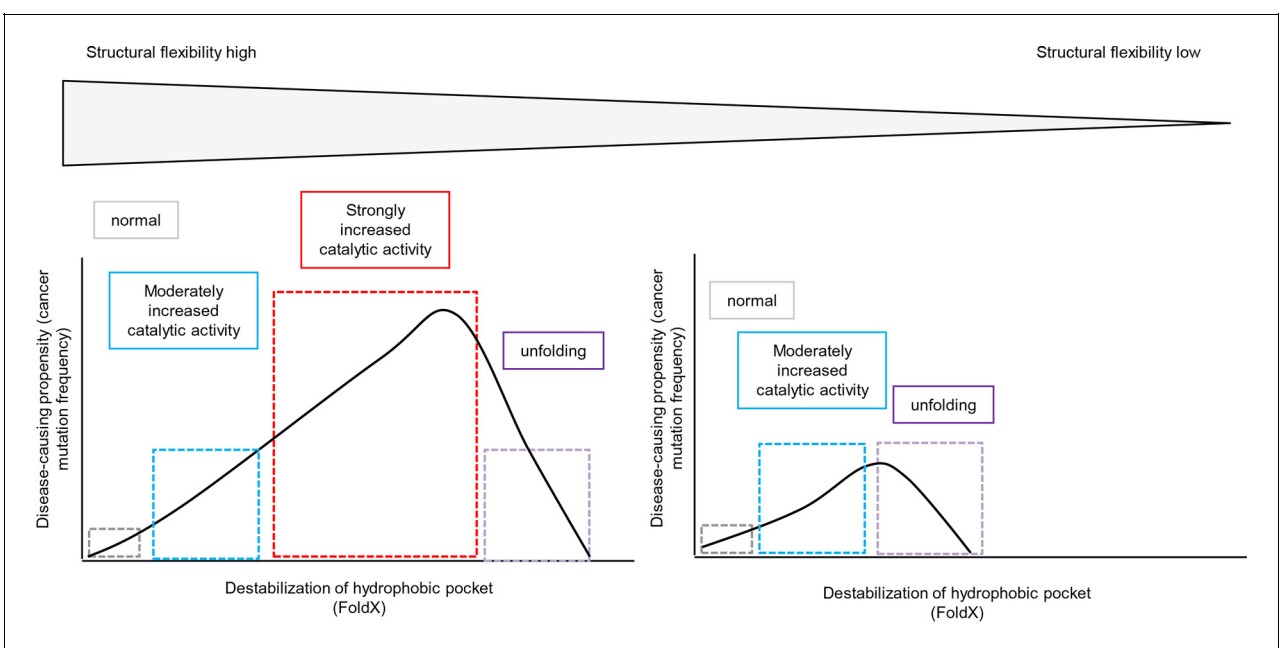

**Figure 6.** Schematic diagram depicting the relationship between structural flexibility, destabilization of the hydrophobic pocket, and cancer frequencies. The effect of a mutation on folding depends on the structural flexibility of the respective hydrophobic pocket where the mutated amino acid is located. In a region with higher structural flexibility destabilization in the hydrophobic pocket will cause activation of the kinase and still allows proper folding. Thus, the disease-causing propensity (cancer mutation frequency) will increase with increasing destabilization of the hydrophobic pocket. In contrast, mutations in structurally rigid areas of the hydrophobic pocket will only slightly increase the catalytic activity with increasing destabilization, and will then cause unfolding.

respective hydrophobic pocket where the mutated amino acid is located (*Figure 6*, left side). Position Val600 is in a region where the destabilization of the hydrophobic pocket causes activation of the kinase as structural flexibility still allows for proper folding. In contrast, those mutations in structurally rigid areas of the hydrophobic pocket only slightly increase the catalytic activity while increasing destabilization, and thus rapidly cause unfolding of BRAF (*Figure 6*, right side).

Altogether, we propose that whether or not BRAF-activating mutations are found in disease depends on the properties of the AS, the associated possibility of disturbing these properties in a single nucleotide substitution and the impact on the stability of the active and inactive conformations. Our results underscore the importance of considering the number of base substitutions required for a given mutation in genome-wide association studies. Rare mutations can be passengers or drivers, depending on the number of base substitutions needed. Additionally, individuals with silent mutations at critical hotspot positions may acquire rare disease mutations infrequently found in cancer. Finally, mutations that both activate and slightly destabilize a protein may be rescued by random fluctuations in the chaperone levels present in a population (*Lehner, 2013*). Consideration of these factors in combination with protein design algorithms may also offer mechanistic explanations of why certain mutations are found at higher frequencies in other oncogenic proteins.

## Materials and methods

### Mutation databases

Somatic BRAF mutations were downloaded from COSMIC (http://cancer.sanger.ac.uk/cancergenome/projects/cosmic/). Germline mutations for BRAF were extracted from OMIM (http://www.omim.org/) and Uniprot (http://www.uniprot.org/).

### Three-dimensional protein structures

Protein structures were retrieved from the Protein Data Bank (http://www.rcsb.org/pdb/home/home.do).

### Protein mutations and stability predictions by FoldX

FoldX (http://foldx.crg.es/) is a computer algorithm that allows interaction energies contributing to the stability of proteins and protein complexes to be calculated (*Guerois et al., 2002*; *Schymkowitz et al., 2005*). For details concerning the force field, please see the description in the online version and in related publications (*Kiel and Serrano, 2009*; *Kiel and Serrano, 2007*; *Rakoczy et al., 2011*). The FoldX algorithm enables predictions of mutational affects for any of the 20 natural amino acids, but not for any backbone changes. Prior to any mutagenesis, we optimized the total energy of the protein using the RepairPDB option of FoldX, which identifies and repairs those residues with bad torsion angles and van der Waals clashes. Mutagenesis was performed using the BuildModel option of FoldX, with five repetitions per mutation. Stabilities were calculated using the Stability command of FoldX, and $\Delta\Delta G$ values were computed by subtracting the energy of the wild-type from that of the mutant.

### The FoldX energy function

The FoldX energy function includes terms that have been found to be important for protein stability. The free energy of unfolding ($\Delta G$) of a target protein is calculated using the equation:

$$\Delta G = Wvdw * \Delta Gvdw + WsolvH * \Delta GsolvH + WsolvP * \Delta GsolvP + \Delta Gwb + \Delta Ghbond + \Delta Gel + \Delta GKon + Wmc * T * \Delta Smc + Wsc * T * \Delta Ssc$$

with:

- $\Delta Gvdw$ as the sum of the van der Waals contributions of all atoms with respect to the same interactions with the solvent
- $\Delta GsolvH$ and $\Delta GsolvP$ as the differences in solvation energy for apolar and polar groups, respectively, when these change from the unfolded to the folded state
- $\Delta Ghbond$ as the free energy difference between the formation of an intra-molecular hydrogen bond compared to inter-molecular hydrogen bond formation (with the solvent)

- ΔGwb as the extra stabilizing free energy provided by a water molecule making more than one hydrogen bond to the protein (water bridges) that cannot be taken into account with non-explicit solvent approximations
- ΔGel as the electrostatic contribution of charged groups, including the helix dipole
- T * ΔSsc as the entropic cost of fixing the backbone in the folded state
- ΔSsc as the entropic cost of fixing a side chain in a particular conformation

If interaction energies between complexes are calculated, two additional terms are needed:

- ΔGKon as the effect of electrostatic interactions on the association constant kon (this applies only to the subunit binding energies)
- ΔStr as the loss of translational and rotational entropy that ensues upon formation of the complex. The latter term cancels out when we are looking at the effect of point mutations on complexes.

## Random forest predictions

Random forest (*Breimann, 2001*) construction and predictions were performed using the package 'randomForest' for R (*R Development Core Team, 2008*) . Two sets of 100 random forests each were constructed. In one set, the V600E mutant was always included in the training set of the classifiers, while in the other it was not. Random forests used ~70% of the samples for the training, with the remaining ~30% was used for performance testing. All random forests were trained with the same parameters. The number of trees was set to 40, as a further increase did not improve the performance of the predictor. The number of variables randomly sampled as candidates at each split of the trees was set to four. To assess the significance of each of the features used in the random forest and how they contribute to the prediction outcome, we determined the importance of each of them. This value is computed by calculating the total decrease in node impurities when splitting on a certain variable. This means that every time a specific variable is used for a split in any of the trees in the forest, the decrease in the impurity of the child nodes, respect to the parent node, is calculated. In regression random forests, this is done by calculating the residual sum of squares, comparing the predicted value of the forest with the real value, for each of the samples in the training. It is expected that the residual sum of squares decreases at each split, thus improving the tree. The larger the decrease, the better the split, and thus the variable used is considered more important. For each variable, the decrease in the node impurity is calculated every time it is used for a split in any of the trees, and the values are added to determine the importance of this variable. The features that contribute most to the random forest prediction will have larger importance values.

## Cloning of wild-type and mutant BRAF

BRAF complementary DNA was cloned into pDEST/N-SF-TAP v1 with N-terminal Strep and Flag tags (provided by Dr. Gloeckner and Dr. Ueffing, HelmHoltz Zentrum Muenchen; (*Gloeckner et al. [2007]*) and fully sequenced. Single amino acid mutations were introduced with the QuikChange site-directed mutagenesis kit (Stratagene) using pDEST/N-SF-TAP BRAF as a template.

## Cell culture, transfection, and Western blot analysis of wild-type and mutant BRAF

HEK293 cells were cultured in Dulbecco's modified Eagle's medium (Gibco) supplemented with L-glutamine and 10% (v/v) heat-inactivated fetal calf serum (here, normal growth medium). For each seeding-transfection-(stimulation)-lysis experiment, HEK293 cells were seeded on 35-mm dishes and transfected after 24 hr (at 80% confluence) with 1 μg of BRAF plasmid, using Lipofectamine 2000 (Invitrogen, Thermo Fisher Scientific, Waltham,Massachusetts, USA) according to the manufacturer's instructions. After 24 hr, cells were washed twice with PBS and resuspended in 200 μl of lysis buffer (0.1% SDS, 25 mM Tris [pH 7.8], 1:1000 protease inhibitor cocktail 1 and 2 [Sigma]). For EGF stimulation experiments, cells were transfected (in serum-free medium) and then, after 1 day, stimulated with 50 ng of EGF or HRG, in 3 ml, for the indicated times, washed with PBS and lysed as above. To fractionate cells into soluble and insoluble fractions, cells were first lysed in hypertonic lysis buffer (20 mM Tris pH 7.5, 5 mM $MgCl_2$, 5 mM $CaCl_2$, 1 mM DTT, 1 mM EDTA, 1:1000 protease inhibitor cocktails 1 and 2 [Sigma]), sonicated for 5 min, and centrifuged for 5 min at 3000 rpm, after which

the supernatant was removed ('soluble fraction'). The pellet was resuspended in SDS lysis buffer ('insoluble fraction'). Cell lysates were loaded for Western blot analysis. Blots were incubated with an enhanced chemiluminescence reagent (SuperSignal West Femto, Thermo 34096) and visualized with a LAS-3000 imager (Fujifilm Co.). Two to three biological sample replicates were generated in each seeding-transfection-lysis experiment and analyzed on the identical Western blot ('biological replicates performed at the same day'). Up to eight different seeding-transfection-lysis experiments were performed ('biological replicates performed at different days'). The intensity of protein bands for MEK-p and flag (for total BRAF levels) was quantified with ImageJ. MEK-p levels were normalized by total BRAF levels (using the flag antibody). To compare biological replicates performed at different days MEK-p/BRAF total intensities were referenced to WT (=100%). While the relative intensity changes between WT and mutants always followed the same trend in all biological replicates performed at different days, the quantitative intensity spread could vary (e.g. for V600E between 180% to 400% compared to WT). To compare intensities from Western blots from different days, we averaged experiments that had a similar intensity spread. The following antibodies were used for Western blotting: Flag (Sigma, F1804), phospho-BRAF Ser445 (Cell Signaling, #2330), phospho-MEK Ser217 and Ser221 (Cell Signaling, #9121), β-actin (Thermo, MA5-15739), and total BRAF (SIGMA, HPA001328).

## Acknowledgements

We thank Walter Kolch and Edgardo Ferran for valuable discussions and Tony Ferrar for critical manuscript revision and language editing (http://theeditorsite.com). This work was funded by the EU (PRIMES under grant agreement number FP7-HEALTH-F4-2011-278568), the Spanish Ministerio de Economía y Competitividad, Plan Nacional BIO2012-39754, and the European Fund for Regional Development. We acknowledge support of the Spanish Ministerio de Economía y Competitividad for the 'Centro de Excelencia Severo Ochoa 2013-2017' (SEV-2012-0208).

# Additional information

### Funding

| Funder | Grant reference number | Author |
| --- | --- | --- |
| European Commission | PRIMES FP7-HEALTH-F4-2011-278568 | Luis Serrano |

The funders had no role in study design, data collection and interpretation, or the decision to submit the work for publication.

### Author contributions

CK, Designed the study and wrote the paper, Performed the bioinformatics analyses and energy calculations of BRAF mutants, Carried out the experiments and analyzed the data, Approved the final version to be published, Conception and design, Acquisition of data, Analysis and interpretation of data, Drafting or revising the article; HB, Carried out the experiments and analyzed the data, Revised critically the article, Approved the final version to be published, Acquisition of data, Analysis and interpretation of data, Drafting or revising the article; VLR, Implemented the random forest analysis and interpreted the data, Revised critically the article, Approved the final version to be published, Acquisition of data, Analysis and interpretation of data, Drafting or revising the article; LS, Designed the study and wrote the paper, Performed the bioinformatics analyses and energy calculations of BRAF mutants, Approved the final version to be published, Conception and design, Analysis and interpretation of data, Drafting or revising the article

### Author ORCIDs

Veronica Lloréns-Rico, http://orcid.org/0000-0002-0860-5990

## Additional files

### Supplementary files

• Supplementary file 1. Summary of structure-energy and statistical properties of mutations in the hydrophobic pocket of BRAF and parameters used for random forest analyses. This table summarizes on sheet 1 for all mutations in the hydrophobic pocket the mutation frequencies in cancer, the average energies predicted using FoldX using active or inactive BRAF template structures, the respective B-factors, the presence of amino acid residues in solved BRAF X-ray structures, the nucleotide substitution frequency, the hydrophobic solvation energy, the change in codon usage, and the energetic contribution of a salt bridge predicted to stabilize the active conformation. Sheet 2 summarizes for all mutations in the hydrophobic pocket the values used to for the seven parameters evaluated in the random forest analyses.

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
