## [Decision Letter]

Thank you for submitting your work entitled "The yin-yang of kinase activation and unfolding explains the peculiarity of Val600 in the activation segment of BRAF" for consideration by *eLife*. Your article has been favorably evaluated by Sean Morrison (Senior editor), a Reviewing editor (Yibing Shan), and two peer reviewers.

The reviewers have discussed the reviews with one another and the Reviewing Editor has drafted this decision to help you prepare a revised submission.

The manuscript attempts to explain why certain BRAF mutations, V600E in particular, occur in cancer cells at much higher frequency than others. Through energetic analyses of the effects of mutations on the overall fold stability of BRAF kinase and the mutations' impact to the catalytically active conformation using FoldX, an empirical structure-based energetic function, the authors showed that the high occurrence of V600E is rooted in the unique combination of promoting the catalytically active kinase conformation and maintaining the overall fold stability of the kinase domain by this mutation. It is further shown that the high occurrence of V600E is in part attributed to the fact that the mutation involves only a single nucleotide change in the codon, as V600D and V600K mutations, which are shown to be similar to V600E in terms of energetics and kinase activity, are much rarer for their requirement of two nucleotide in the codon. This work underscores the balance between functional alteration and impact to the overall fold stability of the protein behind a high-occurrence driver oncogenic mutation.

Summary:

Overall, the reviewers found this work interesting and saw the relevance of this work to cancer biology and potentially, to drug discovery. The reviewers, however, raised a number of concerns and made a number of suggestions to help strengthen this work. In particular, our reviewers raised important questions concerning the quantification of the Western Blot results. In terms of language and details of the figures, the manuscript is somewhat under-polished and not close to a near publishable condition. Some substantial effort on that front will make the manuscript more acceptable to *eLife*.

Essential revisions:

1) How are MEK-p levels normalized? In Figure 4, the ratio of MEKp for WT BRAF is around 1 (bar graph) but the western blot images seem to show much weaker bands for MEKp than for BRAFtotal. Given the quantitative nature of the study, it is important to be convincing in the quantification of Western Blots results.

2) Another main concern has to do with the lack of details of the FoldX analyses. It is not clear exactly what energies are calculated and how they are calculated. For example, more explanation is needed for the data presented in Figure 4 or Figure 5. A basic description of FoldX should be included in the main text that is accessible to a non-computational scientist. In the last sentence of the subsection “A quantitative measure for the destabilization of the hydrophobic pocket using structure-based energy calculations”: What is special about the threshold FoldX energy of 0.8 kcal/mol? Please explain how the threshold is set. Why is only the average free energy of the inactive state corrected for flexibility and not the active state in table 1? Some additional details of the correction are necessary.

3) The energies at the activation segment loop were corrected to take into account its high flexibility and assuming that "mutations at protein regions of high flexibility will have less impact.…". The exact details of the correction and their effect on the energy differences are not clear and this statement needs to be explained in greater details. This is particularly so in the light of the recent paper by Lu et al., Anatomy of protein disorder, flexibility and disease-related mutations, Front Mol Biosci 2015, 2, 47, which discussed the link between the disorder of the loop and the V600E mutation in BRAF.

4) In the subsection “Experimental analysis of mutations in the hydrophobic pocket predicted to disturb protein folding”, first paragraph: Where is the correlation plot between ΔΔG and the soluble:insoluble ratio that is mentioned? This figure seems to be absent.

5) The discussion in the second paragraph of the Introduction implies that only the open cleft state allows key kinase catalytic residues to change conformations. Inactive and active kinase states are not necessarily correlated with open and closed states. The catalytic residues can be in active or inactive conformations in both open and closed states. Furthermore, it should be made clear that activation requires specific conformational changes involving the catalytic residues.

---

## [Author Response]

Essential revisions:

*1) How are MEK-p levels normalized? In Figure 4, the ratio of MEKp for WT BRAF is around 1 (bar graph) but the western blot images seem to show much weaker bands for MEKp than for BRAFtotal. Given the quantitative nature of the study, it is important to be convincing in the quantification of Western Blots results.*

We apologize for this. There were some inconsistencies in the normalization of MEK-P. We have now analysed all blots in the same way, normalizing MEK-P by total BRAF (using the flag antibody). To be able to compare technical and biological replicates performed on different days we then referenced all intensities to the WT (=100%). We added a section to the Methods explaining the analysis and the number of replicates in detail.

We now explain in the figure legend how many biological replicates and technical replicates were used. For the Western blots to be even more convincing, we also show additional Western blots with biological and technical replicates in the new figure supplements: Figure 3—figure supplement 3, Figure 3—figure supplement 4, Figure 4—figure supplement 1, and Figure 4—figure supplement 5.

*2) Another main concern has to do with the lack of details of the FoldX analyses. It is not clear exactly what energies are calculated and how they are calculated. For example, more explanation is needed for the data presented in*
Figure 4
*or*
Figure 5*. A basic description of FoldX should be included in the main text that is accessible to a non-computational scientist. In the last sentence of the subsection “A quantitative measure for the destabilization of the hydrophobic pocket using structure-based energy calculations”: What is special about the threshold FoldX energy of 0.8 kcal/mol? Please explain how the threshold is set. Why is only the average free energy of the inactive state corrected for flexibility and not the active state in* table 1*? Some additional details of the correction are necessary.*

We now explain FoldX better in the main text (results) with the help of an additional supplemental figure (Figure 1—figure supplement 3). We explain the basic molecular forces that contribute to protein folding and that these energies are integrated in the FoldX force field. We then explain how the sidechain modelling of FoldX is used to generate mutant structures and recalculate the energies using the FoldX force field (to result in DDG BRAF Mut-WT). Next, we explain the whole pipeline in terms of modelling mutants of BRAF in the hydrophobic pocket. Finally, we explain how to interpret the data when folding is affected or, alternatively, if the flexible loop is affected, how this will impact the loss of autoinhibitory interactions, but not affect the overall folding.

The threshold of 0.8 kcal/mol represents twice the standard deviation performed using energy calculations with the FoldX force field. We apologize for the lack of explanation and we added this information to the main text.

Three active structures are available. In all these, the residues are solved until residue 600, and therefore no correction should be applied. When we analysed the B-factor of the structure with the AS loop solved (4MNE), we found that B-factors are small until positionVal600, and then high for the remaining positions in the loop. However, as there was no significant destabilization predicted by FoldX for mutations in that loop using the active-like 4MNE, we didn’t apply a correction.

We added the following sentence to the main text: “This correction was not applied to the active-like structures because for these three active structures residues were solved only until position 600. The only available structure for which the loop had been solved (4MNE), had a high B-factor from position 601 onwards, but as there was no significant destabilization seen by FoldX, no correction was applied.”

*3) The energies at the activation segment loop were corrected to take into account its high flexibility and assuming that "mutations at protein regions of high flexibility will have less impact.*…

*". The exact details of the correction and their effect on the energy differences are not clear and this statement needs to be explained in greater details. This is particularly so in the light of the recent paper by Lu et al., Anatomy of protein disorder, flexibility and disease-related mutations, Front Mol Biosci 2015, 2, 47, which discussed the link between the disorder of the loop and the V600E mutation in BRAF.*

We apologize for not having explained this better. There are three types of energies: (i) mutations that destabilize the inactive conformation (using predictions from inactive structures) and affect folding (DDG BRAF_inactive), (ii) mutations that destabilize the active conformation (using predictions from active structures) and affect folding (DDG BRAF_active), and (iii) mutations in the AS loop of inactive structures that will favour its release and cause kinase activation (DDG BRAF_inactive_loop). The folding energies of the inactive structures (DDG BRAF_inactive) have been corrected at regions of high flexibility (such as the AS loop). These mutations will have less impact on unfolding BRAF as structural flexibility still allows proper folding. We now mention in this respect the Lu et al. and Marino et al. papers. “Position Val600 is moderately flexible (70% solved in X-ray structures). This confirms previous predictions that the AS loop belongs to a region within the kinase domain (intra domain region) that has a large tendency to be disordered (Lu et al., 2015). Also, previous enhanced-sampling structure-based computational simulations proposed that the AS exhibited a significant tendency to switch from the ordered to unstructured conformation (Marino et al., 2015).”

*4) In the subsection “Experimental analysis of mutations in the hydrophobic pocket predicted to disturb protein folding”, first paragraph: Where is the correlation plot between ΔΔG and the soluble:insoluble ratio that is mentioned? This figure seems to be absent.*

We apologize for this. We have now added this plot for comparing the values in Figure 5 with Figure 5 (new Figure 4—figure supplement 2).

*5) The discussion in the second paragraph of the Introduction implies that only the open cleft state allows key kinase catalytic residues to change conformations. Inactive and active kinase states are not necessarily correlated with open and closed states. The catalytic residues can be in active or inactive conformations in both open and closed states. Furthermore, it should be made clear that activation requires specific conformational changes involving the catalytic residues.*

We have deleted the second part of the sentence (‘whereby the open form allows catalytic residues to move into the active site (DFG motif and catalytic loop’): “The two lobes, which are spatially connected through the AS, can move relative to each other to open or close the cleft.” We added ‘catalytic’ in front of the Asp: ‘Phosphorylation in the AS causes structural rearrangements of the AS, the aC helix and the phosphate-binding loop, which reorients the *catalytic* Asp of the DFG motif in a catalysis-competent orientation causing BRAF to be active’.